# Antioxidant Activities of Quercetin and Its Complexes for Medicinal Application

**DOI:** 10.3390/molecules24061123

**Published:** 2019-03-21

**Authors:** Dong Xu, Meng-Jiao Hu, Yan-Qiu Wang, Yuan-Lu Cui

**Affiliations:** Research Center of Traditional Chinese Medicine, Tianjin University of Traditional Chinese Medicine, Tianjin 300193, China; dongxu418@163.com (D.X.); humengjiao0626@163.com (M.-J.H.); yqwang1994@163.com (Y.-Q.W.)

**Keywords:** quercetin, polyphenol, antioxidant activity, oxidative stress, complex

## Abstract

Quercetin is a bioactive compound that is widely used in botanical medicine and traditional Chinese medicine due to its potent antioxidant activity. In recent years, antioxidant activities of quercetin have been studied extensively, including its effects on glutathione (GSH), enzymatic activity, signal transduction pathways, and reactive oxygen species (ROS) caused by environmental and toxicological factors. Chemical studies on quercetin have mainly focused on the antioxidant activity of its metal ion complexes and complex ions. In this review, we highlight the recent advances in the antioxidant activities, chemical research, and medicinal application of quercetin.

## 1. Introduction

Quercetin (Figure 1a,b) is a polyphenolic flavonoid compound [1]. It is abundantly present in kales, onions, berries, apples, red grapes, broccoli, and cherries, as well as tea and red wine [2,3]. Modern studies have shown that quercetin prevents various diseases, such as osteoporosis, some forms of cancer, tumors, and lung and cardiovascular diseases. The antioxidant effects of quercetin play a significant role in the prevention and treatment of such diseases [4]. Moreover, owing to its high solubility and bioavailability, quercetin may also exhibit strong antioxidant activity after forming a complex or combining to form some novel preparations used for human health care [5,6,7,8]. At the same time, according to the bibliometric analysis results based on the Web of Science database (Figure 1c), the antioxidant property of quercetin has become a research hotspot [9,10]. Yet, there have been few reviews and summaries focusing on the antioxidant activity of quercetin in recent years [11,12]. Therefore, this paper discusses the antioxidant effects of quercetin from two aspects, biological activity and chemical research, and provides a guideline for the research direction regarding the antioxidant effects of quercetin. In addition, this review describes the application of antioxidants in the medicinal field. This review aims to provide some guidance and reference for future antioxidant research on quercetin.

## 2. Antioxidant Activity of Quercetin In Vivo

The antioxidant activity of quercetin is mainly manifested through its effect on glutathione (GSH), enzymatic activity, signal transduction pathways, and reactive oxygen species (ROS) caused by environmental and toxicological factors. Quercetin shows a strong antioxidant activity by maintaining oxidative balance.

### 2.1. Direct Effects of Quercetin on GSH

Quercetin increases the body’s antioxidant capacity by regulating levels of GSH. This is because, once oxygen free radicals are generated in the body, superoxide dismutase (SOD) quickly captures O^2-^ and transforms it into H_2_O_2_. This enzyme further catalyzes the decomposition of H_2_O_2_ to the non-toxic H_2_O. This reaction requires GSH as a hydrogen donor. Animal and cell studies found that quercetin induces GSH synthesis [13,14]. It was also found that the application of quercetin therapy in renal ischemia/reperfusion (I/R) increased GSH levels, an effect that enhanced the antioxidant capacity of rats [15]. When quercetin is applied at high doses, the dynamic balance of GSH (under the action of GSH peroxidase) is affected; H_2_O_2_ is converted to H_2_O and GSH is oxidized to GSSG (oxidized glutathione disulfide). GSH reductase catalyzes the reduction of GSSG in the liver and red blood cells (by providing H) to form GSH. Thus, the dynamic balance of GSH is produced, which may cause the inhibition of GSH levels in low doses. This inhibition of quercetin on GSH at the level of 0.5% is reported in the literature [16].

### 2.2. Effects of Quercetin on Enzymatic Activity

The –OH groups on the side phenyl ring of quercetin are bound to important amino acid residues at the active site of two enzymes. In this way, it has a stronger inhibitory effect against key enzymes acetylcholinesterase (AChE) and butyrylcholinesterase (BChE), which are associated with oxidative properties [17]. Quercetin has been shown to alleviate the decline of manganese-induced antioxidant enzyme activity, the increase of AChE activity, hydrogen peroxide generation, and lipid peroxidation levels in rats, thereby preventing manganese poisoning [18].

It was reported that pretreatment with quercetin significantly enhanced the expression levels of endogenous antioxidant enzymes such as Cu/Zn SOD, Mn SOD, catalase (CAT), and GSH peroxidase in the hippocampal CA1 pyramidal neurons of animals suffering from ischemic injury. This indicates that it strongly protects the hippocampal area CA1 pyramidal neurons from I/R injury. Thus, quercetin may be a potential neuroprotective agent for transient ischemia [19]. 

In addition, as one of the most metabolically active tissues of the body, bone undergoes a continuous and complex remodeling process throughout life. In particular, osteoblasts, which are derived from osteoprogenitor cells produced by self-renewing, pluripotent stem cells, play a critical role in this cycle. The primary function of osteoblasts is to generate a new bone matrix and, together with osteocytes, support the bone structure itself. Damage to osteoblasts can therefore result in several dysfunctions. Smokers have low bone mass and stability. It was reported that the application of quercetin can promote fracture healing in smokers by removing free radicals and upregulating the expression of heme-oxygenase- (HO-) 1 and superoxide-dismutase- (SOD-) 1, which protects primary human osteoblasts exposed to cigarette smoke [20].

Quercetin has also been shown to prevent heart damage by clearing oxygen-free radicals caused by lipopolysaccharide (LPS)-induced endotoxemia. LPS induces histopathological and biochemical damages to the myocardium in endotoxemia model. In a rat model experiment, rats treated with LPS showed a significant increase in the malondialdehyde (MDA) level in tissues and a decrease in SOD and CAT activity in heart tissues. In contrast, quercetin treatment increased the levels of SOD and CAT and reduced the level of MDA after LPS induction, suggesting that quercetin enhanced the antioxidant defense system [21].

### 2.3. Effects of Quercetin On Signal Transduction Pathways

Quercetin has several effects on various signal transduction pathways, such as activating, inhibiting, upregulating, or downregulating many molecules of the body. In this way, quercetin can improve the antioxidant state of the body and repair injury such as spinal cord injury, atherosclerosis, and lead or cadmium toxicity. Figure 2 shows the antioxidant signal pathways regulated by quercetin.

By influencing signal transduction pathways, quercetin modulates the enzymes or antioxidant substances that enhance antioxidant properties, thereby preventing disease development. Studies have shown that the protective mechanism of quercetin against acute spinal cord injury is mediated by its inhibitory effect on the p38MAPK/iNOS signaling pathway, the downregulation of MDA levels, and the upregulation of SOD activity to promote antioxidant activity [27]. In psoriasis, it was found that quercetin promotes disease recovery by downregulating the expression of NIK and NF-κB including IKK and RelB, and upregulating the expression of TRAF3. It also increases the activity of GSH, CAT, and SOD, and decreases MDA levels in skin tissues induced by imiquimod (IMQ), which collectively enhance the body’s antioxidant performance [28]. 

Several enzymes regulate signaling pathways to enhance antioxidant activities, such as oxidative stress protection. Quercetin reversed the oxLDL-induced decrease in AMPK activation and oxLDL-induced increase in NADPH oxidase expression, thereby maintaining AKT/eNOS function and suppressing NF-κB signal transduction to combat atherosclerosis [29]. Moreover, it can promote oxidation or enhance antioxidant capacity by modulating signal pathways. For example, quercetin controls the development of atherosclerosis induced by a high-fructose diet by inhibiting ROS and enhancing PI3K/AKT. It also promotes the functional recovery of moving medium after cerebral ischemia by promoting antioxidant signal transduction, increasing resistance to apoptosis, and transforming the TGFβ-2/PI3K/AKT pathway [30,31,32]. In humans, quercetin can also prevent or treat damage or toxicity by directly enhancing antioxidant properties through signal transduction pathways, as shown in Table 1.

### 2.4. Effects of Quercetin on ROS Caused by Environmental and Toxicological Factors

Oxidative damage is mostly caused by ROS. Quercetin can remove ROS, thereby resisting oxidative damage such as respiratory damage, ultraviolet radiation b (UVB) skin lesions caused by radiation, and oxidative damage induced by paraquat, as well as sperm change associated with ROS and oxidative damage of gastric epithelial cells. 

Respiratory damage is caused by exposure to fine particulate matter (PM2.5) in the environment, leading to the cellular activity of 16HBE cells, increased production of ROS, and the inhibition of the expression of mitochondrial genes. It has been recognized that quercetin may stimulate 16HBE cells to repair oxidative damage after exposure to PM2.5 by regulating ROS production and anti-inflammatory processes [44].

Human skin is the body’s largest organ that can resist all kinds of environmental damage. However, UVB induces a transient increase in ROS and an imbalance of endogenous antioxidant systems, thereby increasing the level of free radicals and inflammation, which affect cellular processes. Studies have shown that quercetin prevents UVB-induced radiation damage by removing ROS and strengthening the cell membrane and mitochondrion against ROS-induced damage. In addition, it also inhibits cell membrane mobility and mitochondrial membrane depolarization. Therefore, the consumption of quercetin also inhibits this imbalance and is used to prevent UVB-induced skin damage [45,46].

Quercetin prevents oxidative damage induced by paraquat by reducing the ROS levels and increasing the total GSH levels. Other studies have shown that quercetin can alleviate oxidative stress by modulating the expression of antioxidant-related genes in A549 cells [47]. Gastric epithelial injury caused by ROS such as H_2_O_2_ can be suppressed by quercetin treatment due to its protective effects on gastric epithelial GES-1 cells. In these cells, quercetin prevents oxidative damage and inhibits the production of ROS during acute gastric mucosal injury in mice [48]. Moreover, since quercetin has a strong scavenging capacity for ROS, it also protects sperm from ROS and maintains the function of male germ cells [49].

On the other hand, quercetin inhibits oxidative stress, thereby preventing antioxidant damage. Oxidative stress is caused by the imbalance between oxidants and antioxidants in the body, and it tends to be oxidized. Once oxidized, it results in neutrophil inflammatory infiltration, high protease secretion, and other oxidative intermediates. Quercetin inhibits oxidative stress by regulating the balance between oxidant and antioxidant effects. In various experimental studies, quercetin suppressed radiation-induced brain damage in rats, oxidative damage in rats induced by acrylamide, nerve damage in retinas of diabetic rats, as well as neurodegenerative diseases and oxidative stress induced by cadmium fluoride. By modulating the antioxidant levels, quercetin protects the brain, nerves, or other cells in the body from damage caused by oxidation [50,51,52,53,54].

Ionizing radiation induces various types of damage by increasing free radical formation or increasing cell damage and cell death due to ROS. Quercetin effectively protects cells from genetic toxicity and damage induced by radiation by scavenging free radicals and increasing endogenous antioxidant levels. Bioflavonoids act as reducing agents for hydrogen or electronic agents, which inhibit or reduce free radical toxicity and enhance antioxidant properties in the body, thereby providing protection against radiation [55,56,57].

Quercetin exerts antioxidant and hepatoprotective effects against acute liver injury in mice induced by tertiary butyl hydrogen peroxide. This is attributed to its strong antioxidant effect and free radical scavenging effect. It inhibits lipid peroxidation and increases antioxidant activity and thus can be an effective treatment for oxidative liver injury [58]. Quercetin has been found to directly remove ROS and hydroxyl radicals in hypoxia and restore endogenous redox homeostasis by increasing glutathione levels and removing free radical enzyme systems. In this way, it reverses hypoxia-induced memory impairment by reducing oxidative stress-induced neurodegeneration in the hippocampus [59].

Quercetin removes free radicals and strengthens antioxidant defense systems in the body. Thus, quercetin can suppress oxidative stress including the production of ROS induced by nicotine to treat diseases such as tobacco addiction [60].

## 3. Chemical Studies on the Antioxidant Activity of Quercetin

Due to the poor water solubility and low bioavailability (5.3%) of quercetin, several studies have been performed to modify its structure to increase its water solubility and bioavailability, and thus enhance its antioxidant activity [61].

The modification process of quercetin is generally divided into two types—namely, the derivation of quercetin or recombination with other active groups. The former changes the structure of quercetin and improves its solubility through derivation, while the latter produces a synergistic effect based on the properties of active groups and quercetin, such as metal complexes of quercetin. Moreover, the bioactivity and pharmacological action of quercetin are significantly enhanced after forming complexes with some metal or complex ions. Therefore, many researchers have attempted to improve the antioxidant activity of quercetin using the complex formation method.

### 3.1. Complexes with Metal Ions

Combining quercetin with metal ions improves the reducibility of flavonoids by enabling them to be oxidized by free radicals more easily compared to unmatched flavonoids. Therefore, when complexed with metal ions, quercetin shows excellent antioxidant activity. The scavenging capacity of quercetin combined with vanadium [62], copper [63,64], magnesium [65], iron [66], ruthenium [67], cobalt and cadmium [68], calcium [69], and rare earth elements [70] is stronger compared to pure quercetin, based on the DPPH free radical scavenging test. This implies that the antioxidant activity of quercetin complexes is significantly higher than that of pure quercetin. Most of these complexes have been applied in medicine. For instance, the vanadium quercetin complex weakens mammary cancer by regulating the p53 and Akt/mTOR pathways and downregulating cellular proliferation together with increasing apoptosis events. The ruthenium–quercetin complex induces apoptosis in colon cancer cells through a p53-mediated pathway and promotes antiangiogenic activity by inhibiting vascular endothelial growth factor (VEGF). The solid–quercetin rare earth (III) complexes display strong inhibition in tumor cells compared with pure quercetin. Additional studies are required to explore the therapeutic potential and application of other quercetin complexes. However, when quercetin is combined with some metal ions, such as lead [71] and terbium [72], its free radical scavenging and total antioxidant activity are reduced.

### 3.2. Complexes with Complex Ions

A study reported that quercetin does not enhance the activity of SOD, CAT, and GSH-PX in ARPE-19 cells treated with H_2_O_2_ and does not effectively reduce the amount of ROS and MDA in ARPE-19 cells. However, a quercetin–phospholipid complex significantly increased the activity of these enzymes and significantly reduced ROS and MDA levels. These datasets show that the antioxidant activity of quercetin–phospholipid complexes is higher compared to that of free quercetin. The poor water solubility of quercetin limits its use. Therefore, a quercetin–phospholipid complex was developed to improve its water solubility, enhance its absorption through the gastrointestinal tract, and increase its bioavailability [73,74]. The bioavailability of quercetin has also been increased by structural modification with glucoside–sulfate conjugates. Studies have shown that after oral administration of quercetin, about 93.3% of quercetin is metabolized in the intestinal tract and only 3.1% is metabolized in the liver. In contrast, when its structure is altered by forming glucoside–sulfate conjugates, no significant intestinal liver recirculation is observed for the metabolites of quercetin and its conjugates. This indicates that the bioavailability of free quercetin is improved after the incorporation of conjugates, which further enhances the antioxidant activity of free quercetin.

Some complex ionic complexes, such as glucan–quercetin conjugate [75], calcium phosphate–quercetin nanocomplex (CPQN) [76], and quercetin–germanium nanoparticles [77], have higher antioxidant activity than free quercetin. Similar to quercetin complexes with metal ions, complexes with complex ions have been applied in medicine and other pharmaceutical fields.

## 4. Application of Antioxidant Activity in the Medicinal Field 

Given the clinical importance of oxidative damage, antioxidants are expected to treat some diseases. Therefore, quercetin can be exploited in medicinal field due to its strong antioxidant properties. This basic principle of antioxidant activity of quercetin is shown in Figure 3.

### 4.1. Effects of Quercetin on Tumors

Quercetin has been found to influence malignant tumors such as tumors of epithelial tissue and malignant tumors of interlobular tissue. The term "cancer" generally refers to all malignant tumors, including cancer and sarcoma.

Quercetin has been used in cancer prevention and to prevent the spread of various cancers, such as lung, prostate, liver, breast, colon, and cervical cancers. Its anticancer properties are mediated by various mechanisms involving cell signaling pathways and enzymatic activities that inhibit carcinogens. Here, we discuss the treatment or prevention aspect of quercetin for cancer.

High levels of ROS induce oxidative stress, which in turn causes the over-activation of signal transduction pathways and promotes cell proliferation, as well as survival and metabolic adaptation to the tumor microenvironment. In this way, ROS promotes tumorigenesis. Quercetin regulates both internal and external pathways of ROS-mediated protein kinase C (PKC) signaling. PKC is a key regulator of cell growth and differentiation in mammalian cells and its activation partially depends on ROS signaling. PKC inhibits cell proliferation and survival and induces apoptosis in cancer cells. Quercetin prevents cancer development by upregulating p53, which is the most common inactivated tumor suppressor. It also increases the expression of BAX, a downstream target of p53 and a key pro-apoptotic gene in HepG2 cells [82,83].

In addition, quercetin prevents cancer by modulating oxidative stress markers and antioxidant enzymes. In a previous study, histology and oxidative stress markers such as lipid peroxidation (LPO), H_2_O_2_, and antioxidant GSH level were measured in rats. The result showed that rats treated with carcinogen and testosterone had higher levels of LPO and H_2_O_2_ and lower levels of GSH compared to quercetin-treated rats. This implies that quercetin may be used to target signaling molecules in prostate cancer, which is the second highest cause of cancer-related deaths in men [84]. Other studies have confirmed that the levels of antioxidant enzymes and apoptosis proteins in animals with prostate cancer are increased by treatment with quercetin. Studies have found that insulin-like growth factor receptor 1 (IGFIR), AKT, androgen receptor (AR), and cell proliferation and anti-apoptotic proteins are increased in cancer, but quercetin supplementation normalizes their expression [85]. Moreover, quercetin significantly increases antioxidant enzyme levels, including GSH, SOD, and CAT, and inhibits lipid peroxides, thereby preventing skin cancer induced by 7,12-dimethyl Benz (a) anthracene (DMBA) and croton oil in mice. Histology and enzyme activity tests suggest that oral quercetin in the daily diet may decrease the development of skin cancer [86].

### 4.2. Effects of Quercetin on Heart Diseases

Based on its antioxidant activity, quercetin has a therapeutic effect on cardiovascular diseases. For instance, coronary heart disease may lead to acute myocardial infarction (AMI). A recent study showed that oxidative stress is an important factor involved in the development of AMI. Quercetin significantly decreases MDA content, increases SOD and CAT activity, and regulates anti-inflammatory and anti-apoptosis processes to effectively protect against myocardium injury [87]. In addition, quercetin protects the heart from secondary cardiac dysfunction due to oxidative stress and inflammation. Quercetin significantly attenuates ROS overproduction, decreases trauma-induced damage, increases TNF-α, and prevents Ca^2+^ overload-induced myocardial cell injury. Therefore, quercetin can efficiently prevent injury induced by oxidative stress [88].

### 4.3. Effects of Quercetin on Depression

Depression is a common mental disease that negatively affects human physical and mental health. Chronic stress is connected with depression and anxiety. Several studies have shown that quercetin treatment significantly lowers oxidative and inflammatory stress and prevents neural damage by regulating oxidative stress markers such as TBARS and nitric oxide, antioxidants such as total thiol and catalase, and pro-inflammatory cytokines in the hippocampus [89]. Elsewhere, it has been shown that quercetin has a significant antidepressant effect on the olfactory bulbectomy (OB), an animal model of depression. In a forced swimming test and tail suspension test, quercetin reduced the immobility time, increased the time spent grooming, and increased the SOD activity and lipid hydroperoxide content (LOOH) in the hippocampus, thus providing antidepressant effects [90]. These findings indicate that quercetin prevents stress induced by neurological complications and enhances the total antioxidant activity in the body.

### 4.4. Effects of Quercetin on Diseases Caused by Poisoning

Quercetin has the potential to treat diseases caused by pathogenic factors. Such diseases are often caused by an imbalance between oxidation and antioxidation processes. Quercetin protects the body from urotoxicity induced by cyclophosphamide (CYP) via the inhibition of oxidative stress and the restoration of pro-inflammatory/anti-inflammatory cytokine balance. Compared with normal group, CYP treatment showed a marked reduction in bladder levels of catalase, SOD, and IL-10. However, quercetin prevented urotoxicity by reversing the changes in these biochemical markers and histopathology [91]. By inhibiting CYP 2E1 activity, quercetin reduces the production of ROS and peroxide oxidation, which prevents liver injury in type I diabetes [92]. In addition, quercetin exerts therapeutic effects on diseases induced by manganese [93], ciprofloxacin [94], glycosides from thunder god vine [95], cadmium (Cd) [96], Procarbazine (PCZ) [97], arsenic [98], LPS [99], and cisplatin (Cis.) [100]. Quercetin can treat diseases caused by oxidative damage by increasing the body’s total antioxidant activity and related antioxidant enzymes.

### 4.5. Effects of Quercetin on Other Diseases

Numerous studies have shown that quercetin has the potential to treat other diseases, such as necrotizing enterocolitis (NEC), diabetes, and lung injury. In NEC, quercetin modulates the total antioxidant status by regulating serum levels of MDA and GSH [101]. In type II diabetes, a study found that quercetin intake was inversely related to the prevalence of diabetes [102]. Quercetin has been proposed to be a remedy for idiopathic pulmonary fibrosis (IPF) due to its capacity to restore pulmonary redox balance and suppress inflammation [103]. Quercetin prevents acute lung injury by reducing the levels of oxidative stress markers and increasing antioxidant enzyme activities [104].

## 5. Conclusions

Quercetin is a typical flavonoid that is abundant in fruits and vegetables. Its application in the medicinal field has shown potential to improve human health due to its antioxidant activity in vivo. Some studies show that quercetin can be used as a nutraceutical to offer protection against various diseases. It is effective in the treatment and prevention of human diseases since it influences glutathione, enzymes, signal transduction pathways, and ROS production. However, its application in the pharmaceutical field is limited by its low absorption into the body based on its poor solubility, low bioavailability, poor permeability, and instability. When it forms complexes with metal ions or complex ions, its bioavailability and antioxidant effect are enhanced and strengthened. In addition to quercetin complexes, newer preparations of quercetin have emerged in recent years, including nanoparticles loaded with quercetin [105,106,107], polymeric micelles of quercetin [108], quercetin-loaded mucoadhesive nanoemulsion [109], quercetin-loaded gel [110], and others [111,112]. These preparations improve the solubility and bioavailability of quercetin, which enhances its clinical efficacy and offers new drug formulations for research and development.

Therefore, strategies that improve the solubility and bioavailability of quercetin will potentiate its properties such as its antioxidant and antimicrobial activities. This will contribute to the full exploitation of quercetin as a rich natural drug resource for medicinal uses.

## Figures and Tables

**Figure 1 molecules-24-01123-f001:**
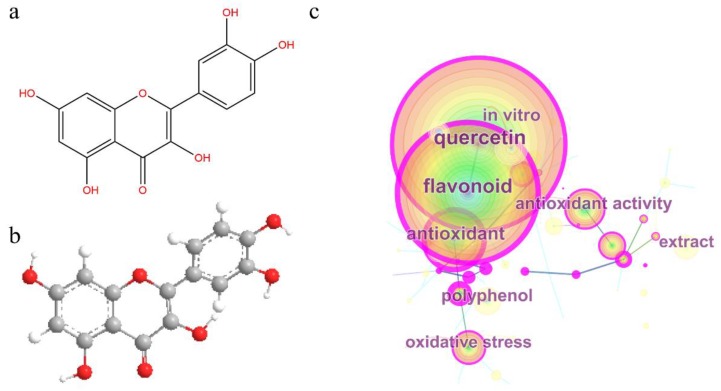
Structure and bibliometric results of quercetin. (**a**) Chemical structure of quercetin. (**b**) 3D conformer of quercetin. (**c**) Co-occurrence map of quercetin. The figure is based on data in the Web of Science (WOS) database ranging from 2000 to 2017, and was drawn by CiteSpace. The diameter of a node represents the number of occurrences of keywords. The larger the diameter, the greater the number of appearances.

**Figure 2 molecules-24-01123-f002:**
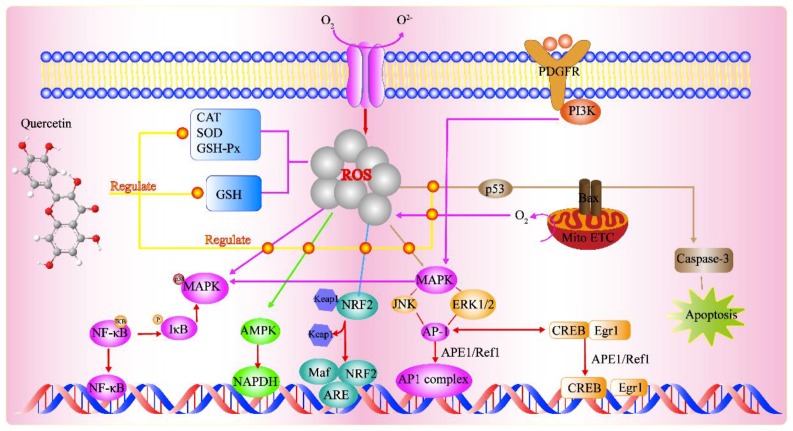
The antioxidant signaling pathway regulated by quercetin. Environmental factors increase the production of reactive oxygen species (ROS). The mitochondrial electron transport chain (mito ETC) is another robust source of intracellular ROS generation. Quercetin can regulate the enzyme-mediated antioxidant defense system and the non-enzyme-dependent antioxidant defense system. It can also regulate signal pathways such as NRFB, AMPK, and MAPK caused by ROS to promote the antioxidant defense system and maintain oxidative balance. ROS in turn enhance the production of APE1/Ref1 and the activation of several signaling events including p53-mediated apoptotic events, MAPK pathways, the NF-E2-related factor (NRF2)-mediated activation of genes containing antioxidant response element (ARE), and NF-κB [22,23,24,25,26].

**Figure 3 molecules-24-01123-f003:**
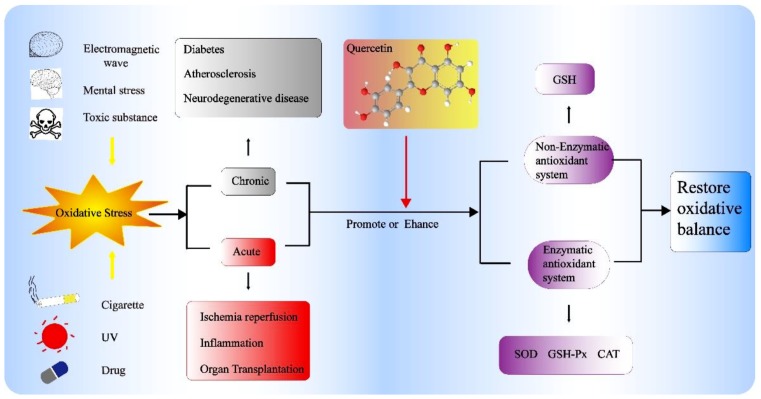
Basic principle of antioxidant activity of quercetin [78,79,80,81].

**Table 1 molecules-24-01123-t001:** Mechanisms of quercetin for treating damage induced by various factors.

Inductive Factors	Damage Name	Protection Mechanisms	Result
LPS/d-GalN	Acute liver injury	Inhibits the activation of NF-κB and MAPK signaling pathways and inhibits the expression of apoptosis-related proteins induced by LPS/d-GalN	Decreases the production of LPS/d-GalN induced by oxidation markers [33]
Toosendanin	Liver toxicity	Induces Nrf2/GCL/GSH antioxidant signal transduction pathways	Increases Nrf2-mediated GCLC/GCLM expression, thereby increasing GSH content in cells [34]
Alcohol	Liver damage	Regulates phosphoinositide 3-kinase/Akt/NF-κB and STAT3 pathways	Enhances the body’s antioxidant, anti-inflammatory, and anti-apoptotic effects [35]
A variety of liver toxins	Liver toxicity	Induces p62 expression and inhibits the binding of Keap1 and Nrf2	Increases the transcription expression of Nrf2-targeted antioxidant genes [36]
Doxorubicin	Heart toxicity	Upregulates Bmi-1 expression to reduce oxidative stress	Reduces DNA damage at ROS levels and maintains cardiomyocyte viability [37]
CCl_4_	Liver damage	Improves antioxidant activity and regulates TLR2/TLR4 and MAPK/NF-κB signaling pathways	Inhibits ROS production in the liver and attenuates CCl_4_-induced oxidative damage [38]
Lead	Liver damage	Reduces oxidative stress in liver, inhibits JNK phosphorylation, and increases PI3K and Akt levels	Effectively inhibits lead-induced endoplasmic reticulum stress [39]
Cadmium	Cerebral cholinergic dysfunction	Reduces the production of ROS and protects the integrity of the line by regulating the protein involved in apoptosis and MAPK signal conduction	Regulates molecular targets involved in the signal conduction of brain cholinergic energy and reduces the neurotoxicity of cadmium [40]
	Malignant cell transformation	Protects BEAS-2B cells from Cr (VI) induction by targeting miR-21-PDCD4 signaling	Reduces ROS production induced by Cr (VI) exposure in BEAS-2B cells [41]
d-lactose	Cognitive impairment and neuron degeneration or loss	Improves the Nrf2-ARE signaling pathway	Decreases free radicals, increases antioxidant enzyme activity, improves overall antioxidant capacity, and slows down aging by improving Nrf2 [42]
Receptor activator for NF-κB ligand	Osteoblast differentiation	Regulates the transcription activities of NF-κB and AP-1	Inhibits the NF-κB and AP-1 mechanism activation [43]

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
