# Peer review of "Antioxidant Activities of Quercetin and Its Complexes for Medicinal Application"

_molecules, 2019, doi:10.3390/molecules24061123_

Round 1
Reviewer 1 Report
This review provided a novel viewpoint on the antioxidative activity of quercetin and its complex, including its medicinal application. The review should make a minor revise before publication:
1. The abbreviation should be given the full name for the first time.
2. Please correct antioxidant to antioxidation in the last sentence of abstract.
3. Please correct Cl4 to Cl4 on page 4.
4. The IKK and RelB were the members of NF-κB, so the statement should be revised in this section on page 4 (line 101-102).
5. What the mean of inhibiting TGFβ2/ PI3K/AKT activation (line 112)? Please improve it.
6. The sentence of “Some studies have shown that quercetin prevents UVB-induced radiation damage by removing ROS and preventing cell membrane and mitochondrion from attacks of ROS and inhibits cell membrane mobility and mitochondrial membrane depolarization.” should be improved (line 127-130).
7. In the “Conclusion”, this section should be summarized without references, please improve it.
8. The manuscript should be improved by the native English speaker.
Author Response
Point 1: The abbreviation should be given the full name for the first time.
Response 1: Modified.
Point 2: Please correct antioxidant to antioxidation in the last sentence of abstract.
Response 2: In this review, we highlight the recent advances in the antioxidant activities, chemical research, and medicinal application of quercetin.
Point 3: Please correct Cl4 to Cl4 on page 4.
Response 3: Modified.
Point 4: The IKK and RelB were the members of NF-κB, so the statement should be revised in this section on page 4 (line 101-102).
Response 4: In psoriasis, it was found that quercetin promotes disease recovery by downregulating the expression of NIK and NF-κB including IKK and RelB, and upregulating the expression of TRAF3.
Point 5: What the mean of inhibiting TGFβ2/ PI3K/AKT activation (line 112)? Please improve it.
Response 5: Transforming TGFβ-2/PI3K/AKT pathway.
Point 6: The sentence of “Some studies have shown that quercetin prevents UVB-induced radiation damage by removing ROS and preventing cell membrane and mitochondrion from attacks of ROS and inhibits cell membrane mobility and mitochondrial membrane depolarization.” should be improved (line 127-130).
Response 6: Studies have shown that quercetin prevents UVB-induced radiation damage by removing ROS and strengthening the cell membrane and mitochondrion against ROS-induced damage. In addition, it also inhibits cell membrane mobility and mitochondrial membrane depolarization.
Point 7: In the “Conclusion”, this section should be summarized without references, please improve it.
Response 7: Modified.
Point 8: The manuscript should be improved by the native English speaker.
Response 8: The manuscript had been improved by the native English speaker.
Reviewer 2 Report
The present manuscript presents a literature review regarding antioxidant activities of quercetin and its complexes and also their possible application in medicine. The topic is very interesting and very well presented. However, there are some corrections that needs to be done. Please find attached pdf of the manuscript with corrections.

Author Response
Some errors had been corrected,depending on the attached pdf of manuscript. At the same time,the manuscript had been improved by the native English speaker